# Eliciting Requirements for a Diabetes Self-Management Application for Underserved Populations: A Multi-Stakeholder Analysis

**DOI:** 10.3390/ijerph19010127

**Published:** 2021-12-23

**Authors:** Samuel Bonet Olivencia, Arjun H. Rao, Alec Smith, Farzan Sasangohar

**Affiliations:** 1Department of Industrial & Systems Engineering, Texas A&M University, College Station, TX 77843, USA; Samuel089@tamu.edu (S.B.O.); Arjun.rao@collins.com (A.H.R.); Alec.smith@tamu.edu (A.S.); 2Center for Outcomes Research, Houston Methodist, Houston, TX 77030, USA

**Keywords:** diabetes mellitus, self-management, blood glucose self-monitoring, mobile applications, medically underserved area, health literacy, telemedicine, disease management

## Abstract

Medically underserved communities have limited access to effective disease management resources in the U.S. Mobile health applications (mHealth apps) offer patients a cost-effective way to monitor and self-manage their condition and to communicate with providers; however, current diabetes self-management apps have rarely included end-users from underserved communities in the design process. This research documents key stakeholder-driven design requirements for a diabetes self-management app for medically underserved patients. Semi-structured survey interviews were carried out on 97 patients with diabetes and 11 healthcare providers from medically underserved counties in South Texas, to elicit perspectives and preferences regarding a diabetes self-management app, and their beliefs regarding such an app’s usage and utility. Patients emphasized the need for accessible educational content and for quick access to guidance on regulating blood sugar, diet, and exercise and physical activity using multimedia rather than textual forms. Healthcare providers indicated that glucose monitoring, educational content, and the graphical visualization of diabetes data were among the top-rated app features. These findings suggest that specific design requirements for the underserved can improve the adoption, usability, and sustainability of such interventions. Designers should consider health literacy and numeracy, linguistic barriers, data visualization, data entry complexity, and information exchange capabilities.

## 1. Introduction

In 2020, over 30 million individuals in the United States suffered from diabetes, most (about 90%) with type 2 diabetes [1]. Rural/medically underserved areas—defined as populations with low access to primary care providers, high infant mortality, high poverty, and/or high elderly population [2]—have shown relatively poor diabetes outcomes compared to the urban/well-served areas [3]. Additionally, type 2 diabetes disproportionately affects people of certain racial and ethnic groups, many of whom may live in areas identified as rural/medically underserved [4], such as Hispanics/Latinx Americans. Recent estimates from the Centers for Disease Control showed that individuals from Hispanic/Latinx American heritage were more likely (17%) to suffer from diabetes than the non-Hispanic White population (8%) [5]. Additionally, data from the U.S. Department of Health and Human Services’ Office of Minority Health revealed that diabetes was among the leading causes for mortality among the non-White population [6].

Effective self-management of diabetes can have a significant impact on health outcomes. Studies have shown that patients who received training in self-management were successful in regulating their blood glucose levels, dietary habits, and glycemic control [7,8,9,10,11]. However, several barriers restrict the ability of underserved patients to execute self-management effectively [12]. These include limited access to timely healthcare services [13], limited financial resources [14], low literacy [15,16,17], and geographic barriers to seeking care from providers outside their community [18]. Therefore, there is a need to investigate methods or interventions that enable self-management by identifying and addressing such barriers systematically.

Telehealth, a type of health information technology, has received special attention in recent years for improving access to health care [19], and for supporting integrated care for chronic diseases by providing patient education and information transfer between patients and providers [20]. Recent advances in mobile health (mHealth) technologies, a modality of telehealth interventions, have shown promise in mitigating barriers related to accessibility. These technologies facilitate the self-management of diabetes, including discreet, cost-efficient, and non-invasive tools for monitoring health conditions, and a reliable platform for interactions between healthcare providers and patients [21,22]. A recent review of 11 mHealth apps for diabetes [23] revealed common features, such as setting reminders, tracking blood glucose and hemoglobin A1c (HbA1c), medication use, physical activity, and weight which support self-management. mHealth technologies, integrated with monitoring technologies, such as glucometers and continuous glucose monitors, have also shown promise in improving healthcare delivery [24]. Additionally, the recent integration of machine learning algorithms and artificial intelligence with mHealth played a vital role in the use of the data collected by these technologies for clinical decision-making [24]. 

While such characteristics make mHealth a promising method to address barriers to self-management in underserved populations, there is limited research documenting guidelines or mHealth design requirements for these populations. Previous research highlights the importance of supporting different languages and cultures for the improved adoption and sustainability in underserved populations. For example, Burner et al [25] and Williams et al. [26] discussed the need for providing basic features, such as educational content [25,26], reminders [26], and user interfaces in Spanish for Hispanic/Latinx users. In addition, glucometer connectivity functionalities [26], and the personalization of messages and content [25] are discussed as features that are important for users in underserved populations; however, such features are typically lacking in those apps available in the market [26]. Low health literacy and eHealth literacy have been identified as potential challenges for the sustained engagement of vulnerable populations with electronic, mobile, and telehealth tools; a systematic review shows that these factors have been underassessed in the published literature about the design of mobile interventions [27]. Additionally, research revealed that paid mobile apps are more likely to integrate strategies to engage low health literate populations, in comparison with free mobile apps [28]; however, cost has been identified as a major concern for people to download and adopt mHealth apps [29], and financial barriers can restrict underserved populations’ self-management of their chronic conditions [14,30]. Age has been another factor highlighted in the literature affecting mHealth usage and adoption. While younger individuals have been identified as more likely to engage with mHealth apps, it is vital to assess design consideration for the elderly population [29]. For example, the use of simple, actionable, and information rich visualizations can help to address some of the design limitations of the low health literate and elderly population [31]. The patients’ intrinsic level of motivation has also been linked to vulnerable the populations’ level of engagement with mobile interventions [27]. Research has shown the need to apply design techniques, such as sequential multiple assignment randomized trials (SMART), to tailor self-monitoring mobile interventions to the patients’ individual level of internal motivation [32]. 

Despite the evidence suggesting users’ preferences for personalized nutritional and health behavior content [32], research [33] highlighted the lack of personalized feedback and significant usability issues, including ease of data entry and integration with patients and electronic health records, suggesting a potential gap in user-centered design (UCD) approaches. Indeed, usability tests on eight mHealth apps for diabetes revealed that more than two-thirds (6/8) were scored by patients as “marginal” or “not acceptable” [23]. This is supported by another study, in which about half of the participants reported stopping their use of mHealth apps due to a high data entry burden and loss of interest, among other factors [29]. UCD has shown promise in fostering user engagement and improving the perception of app effectiveness, with positive impact on sustained behavioral change [34,35]. To our knowledge, only a few attempts have been documented to utilize UCD to inform requirements for diabetes self-management apps (e.g., [36]) and no research has focused on the needs and expectations of the underserved. To address this gap, in this paper, we document the stakeholders’ needs and expectations from a diabetes self-management app, by eliciting feedback from patients with diabetes and providers in several medically underserved areas in the United States. 

## 2. Methods

Semi-structured interviews were conducted with a convenience sample of 97 patients and 11 healthcare providers from several medically underserved counties in South Texas. The interviews with patients were conducted by four nurse educators with graduate degrees in nursing or education during diabetes self-management education sessions, held between 8 April 2019 and 3 May 2019, as part of the Healthy Texas initiative. These education sessions aimed at educating patients with diabetes on practical strategies and tips for incorporating healthy behaviors in their daily activities, including effective nutrition, general health and wellness, the role of physical activity, and ways to mitigate the financial and physical burden of diabetes. Participants were informed about the study at the end of educational sessions and were selected if they met the participation criteria (aged 18+ and had diabetes). The authors FS and AR, who held doctoral degrees in Engineering and had extensive experience in qualitative research, provided these interviewers with training on conducting interviews. Providers were recruited and interviewed by the authors AR, SB, and FS during a diabetes education conference in South Texas. The research group used a booth at the conference exhibition room to recruit the participants. The study team did not establish a relationship with the participants prior to the study and no one other than the interviewers were present at the interview sessions. No potential participant refused or withdrew mid-study, and no repeated interviews were carried out. 

Two interview protocols were developed for patients and providers, respectively, to reduce individual biases and assumptions and to standardize the interviews. Interviews with patients focused on understanding their expectations from a diabetes self-management app. The questions in the interview protocol for patients included topics, such as perceived barriers and limitations for diabetes self-management, the use of technology to manage diabetes, important characteristics in a technology for diabetes self-management, and preferences on features for an app for diabetes self-management. Similarly, interviews with providers focused on their expectations from a self-management app for diabetes both from the patients’ perspective and the type of information or interactions providers expected from such a tool. The questions in the interview protocol for providers included topics, such as perceived barriers and limitations for patients to adopt and app for diabetes self-management, perceived barriers and limitations for providers to monitor patient who have adopted such technology, perceived importance on feature for an app for diabetes self-management, and preferences about data representation and data communication. The interviews took approximately 45 min for both the patients and providers. The patients and providers received a USD 25 or USD 50 gift card, respectively, for participation. The Texas A&M University Review Board reviewed and approved this study (IRB Protocol #IRB2018-1503D) and all participants provided informed consent. 

The interviews were audio recorded and no field notes were made by the interviewers during or after the interviews. A transcription service, Temi, was used to transcribe the audio recorded interviews preceding analysis [37]. The thematic analysis of the interviews was conducted by two coders (AS and AR) [38,39]. The two coders completed the following steps, separately and sequentially, and then met to discuss any discrepancies: code creation, initial coding, and focused coding. The thematic coding process entailed a deeper discussion of the themes and constructs that emerged from the analysis. After coming to a consensus, the themes were discussed with the other authors (SB and FS) and changes were made, as necessary. MAXQDA 12 was used to complete the analysis [40]. AS and SB were doctoral students and had extensive experience in qualitative data analysis.

## 3. Results

### 3.1. Demographics of Participants

#### 3.1.1. Patient Demographics

Table 1 presents the key demographics of the patients. A total of 100 patients participated in the interviews. After cleaning the data, removing incomplete entries, a total of 97 interviews were analyzed. The average age of the participants was 56.07 (SD = 13.10). A vast majority of the participants were Hispanic or Latinx (90%, 87/97). Most of the patients did not have a postsecondary degree, with 73.20% (71/97) of the respondents having either some college (no degree), a high school diploma, or less. Approximately half the respondents (50.51%; 49/97) had a household income of less than USD 30,000. About a fourth of participants reported not having medical insurance (24%; 23/97). A majority of the participants had type 2 diabetes (81%, 79/97). About a fourth of participants were diagnosed with diabetes within a year of the date of their participation in the study (25%, 24/97), and about 39% (38/97) of respondents reported having diabetes for more than 10 years. 

#### 3.1.2. Healthcare Provider Demographics

Eleven healthcare providers serving medically underserved communities in South Texas participated in the interviews. Table 2 presents the key demographics for the healthcare providers interviewed. On average, the physicians sampled had nearly 3 decades (mean = 28.86; SD = 7.75; and range: 10 to 38) of experience in their current roles. Most participants (9/11) practiced family medicine, one practiced general medicine, and one was a pediatric nurse practitioner. Two participants held leadership roles (president/CEO) in their respective organizations.

### 3.2. Participant Interview Themes

Patients were asked to specify features they desired in a diabetes self-management mobile app. A total of 97 participants responded to this question. The analysis of these responses resulted in five superordinate themes: (1) logging and tracking of blood sugar readings; (2) assistance with adopting a healthy lifestyle; (3) integration with the healthcare system; (4) reminders and alerts; and (5) usability and non-invasiveness. Almost 20% of the respondents (19.58%; 19/97) indicated that they did not know what features they would expect in a diabetes self-management app. 

The healthcare providers were asked a series of questions about features they believed would benefit their patients and would improve their practice. The analysis of the responses from 11 physicians resulted in 5 superordinate themes: (1) dietary logs; (2) patient diabetes education; (3) reminders and alerts; (4) information communication and presentation; and (5) patient-related challenges and barriers. These themes and associated subthemes are discussed below. The proportion of participants whose response is captured by a theme or subtheme is presented with percentage (%) and counts (xx/XX). Some subtheme counts do not total 100% because some participants had responses in multiple subthemes.

#### 3.2.1. Functional Requirements Suggested by the Patients

##### Logging and Tracking Blood Sugar Readings

Almost a quarter (24.74%; 24/97) of the patients in our sample expressed the need to be able to track and log their blood sugar readings. Two prominent subthemes were identified from the interviews: (a) logging readings and (b) assistance and insights from the readings.

**Logging Readings**: this subtheme captures the patients’ desire for the app to help them log and recall their blood sugar readings. More than half of the patients (54.17%; 13/24) who expected this feature also highlighted the need to trace back to previous readings to check their well-being.


*“Just to be able to keep track of myself… or tracking my glucose… without having to write it down”*

*—P12*


**Assistance & Insights from the Readings**: some patients (16.67%; 4/24) pointed out the need to understand what the entries mean. Specifically, they indicated the need for graphical interfaces to visualize the trends of their readings. For example, *“keeping track of history…so I can monitor for trends” (P19)*. The participants mentioned familiarity with similar visual trends, such as activity and expected similar visualization for sugar levels. In addition, some of the patients mentioned that descriptive statistics about their parameters would be useful in managing their condition, such *as “the daily average, and the weekly average” (P09)*.

##### Assistance with Adopting a Healthier Lifestyle

About a third (34.02%; 33/97) of the respondents indicated the need for assistance with managing their condition and adopting healthier choices, and demonstrated a willingness to learn about tips and techniques to manage their diabetes. Specifically, their responses were categorized into three subthemes: (a) diet regulation; (b) health tips; and (c) fitness and physical activity.

**Diet Regulation****:** this theme captures the patients’ desire to be provided with information on regulating their eating habits and food intake. Two thirds (66.67%; 22/33) of these participants wanted diet regulation features, including access to a list of the types of foods they can consume to maintain glycemic control. Furthermore, patients also wanted the app to help them to construct and adhere to a diet plan. Finally, some patients (based on the diabetes education they had received) indicated that they could benefit from having a carbohydrate “tracker”.


*“Like maybe like a diet plan, things to do or not to do you know that can lower your sugars if they’re high.”*

*—P21*



*“[…] and a list of dos’ and do not food, you know, like a list, an actual list.”*

*—P07*



*“How many carbs, I can [eat], you know, in, um, like in the mornings […or] at lunchtime I’ll have a sandwich […] I think that’s one of the reasons my diabetes goes up. It scares me, you know, to eat a lot of carbs.”—P16*


**Health Tips****:** patients frequently (39.39%; 13/33) mentioned the need to easily access health-related resources. Although there was an interest in health resources in general, patients were particularly keen on accessing specific tips about nutrition. The participants also mentioned expecting prescriptive tips when presented with abnormal blood sugar values.


*“There’s a lot of things like for your heart and stuff […] there’s a lot of stuff out here that we eat and we’re not supposed to because it’s really damaging ourselves. So, you know some advice […] give us something like that.”*

*—P11*



*“[…] to see, to measure if your sugar is high or low and to explain what things you can do to lower our sugar […]”*

*—P08*


**Fitness and Physical Activity**: several of these patients (15.15%; 5/33) indicated that, while there are several commercial apps for fitness and activity tracking, a fitness module integrated into the diabetes self-management app would be ideal, suggesting the perceived importance of the connection between physical activity and diabetes.

##### Reminders and Alerts

Some patients (11.34%; 11/97) suggested timely alerts or reminders would help them adhere to their medication regimen. Two subthemes emerged from patient responses: (a) reminders and scheduling, and (b) predictive capability.

**Reminders and Scheduling**: several of these patients (72.72%; 8/11) highlighted their busy lifestyle as a reason for forgetting to monitor their blood sugar levels. In addition to being reminded to monitor their health, patients also suggested that a scheduling tool would help them keep track of their appointments.

**Predictive Capability**: some of these patients (36.36%; 4/11) responded that they would like predictive features, such as the early detection of warning signs and monitoring trends, so they can mitigate any problems before they occur.


*“[…] maybe signs to look for, like when you’re going to have maybe an [hypoglycemia] episode, so like warning signs.”*

*—P39*


##### Integration with Healthcare System

A few patients in our sample (3.09%; 3/97) highlighted the need for their diabetes self-management program to be integrated into their overall care system. The patients desired easy communication of diabetic parameters and progress reports with their healthcare provider.


*“[…] being able to send it to the doctor, or bring a recording of the reading. That way they could keep track of it.”*

*—P06*


##### Usability and Non-Invasiveness

Several patients (13.4%; 13/97) desired a system that would be easy to use and non-invasive. This superordinate theme can be categorized into two subthemes: (a) usability and (b) non-invasiveness.

**Usability**: Several patients raised concerns about their experiences with app usability (46.15%; 6/13) and expected an app that was reliable, accurate, and easy to use.

**Non-invasiveness**: this theme captures the patients’ desire for a method of reading sugar levels without having to prick themselves, as commonly required by most glucometers. Most of these patients (53.84%; 7/13) were fatigued by the constant pricking for the blood sugar measurement and desired an app that would display blood sugar readings (potentially from an implantable continuous glucose monitor).


*“Like I said, a feature that would allow you to check your glucose level without [pricking], …, I mean I don’t know if they can make something like that without drawing your blood.”*

*—P14*


#### 3.2.2. Functional Requirements Suggested by Healthcare Providers

##### Dietary Logs

Healthcare providers (72.72%; 8/11) highlighted the importance of a diet/food log to keep track of what patients are consuming and to have patients engage in their treatment. Healthcare providers also suggested that the app should provide immediate feedback to the patients about the calorie density and quality of the food they are ingesting. However, providers cautioned against using food logs in clinical assessments as patients tended to be dishonest in their logs.


*“So, I mean, if they want to write it down, that’s fine […] if you’re assuming perfect compliance and honesty. But my experience is that most patients aren’t completely honest with what they do. So, I guess in the ideal setting, a food log would be great. So, you can go, I see when you have that bowl of ice cream, you know, that wasn’t broccoli, you know, then food log could be really important. So, I guess we could change it.”*

*—S01*



*“Food log with […] input about calorie and everything else. So, it’d be two-way […] Immediate feedback. Get pretty much immediate feedback. If they’re going to go to the trouble of entering in all that food, they need to get, I don’t want it just to be written down, you know, and just stored somewhere and they look it up. They’re going to enter what they’re going to eat in a food log. They need to get immediate feedback about the calories or, and this is on or off their diet or something like that.”*

*—S08*


One physician alluded to the use of image processing and machine learning techniques to analyze a photograph of a plate of food. The results of the analysis provide a breakdown of calorie content and nutritional value.


*“I thought it would be fantastic if a person sets their meal, their plate down, they take a photo of it. And artificial intelligence calculates the, based on the size of the plate, I mean, […], how much potatoes take up, how much the meat takes up. And it calculates […]. We load the fat amount, the protein amount [but] I don’t think they have that yet.”*

*—S04*


##### Patient Diabetes Education

Some providers (27.27%; 3/11) highlighted the low health literacy of patients they treated and encouraged the creation of an educational component in the diabetes self-management app for facilitating communication with educators.


*“We have to give them the information… It’s like a coach. This is the game plan… this is how you throw the ball and all that. [You have] repetition and they get better at it.”*

*—S08*


##### Reminders and Alerts

Most providers (72.72%; 8/11) suggested that providing periodic reminders or alerts about multiple topics, including ingesting medication, diet adherence, activity reminders, and appointments, would benefit patients and help them in self-management.


*“Self-management. So yeah, you get reminders. You got to do that for them. Probably about every two hours… you remind them about if your glucose is too high or too low… They could do a reading… to help them for self-management.”*

*—S02*


Providers also cautioned designers about the tendency for patients to develop alarm fatigue leading to ignored reminders, thus highlighting the need to remind or alert only when appropriate.


*“[…] there are patients who may feel like this is getting a little [annoying], and you’re going to have to see everybody [feels] a little intruded.”*

*—S08*



*“[…] when you start getting emails that are 12 different things on the same subject, you just start going through them and not reading them. And that’s what we’re seeing. They will gloss over them.”*

*—S05*


When discussing reminders, the healthcare providers’ responses could be categorized into two subthemes: (a) medication intake recall and (b) activity reminder

**Medication Intake Recall**: several providers (50%; 4/8) indicated that some of their patients had trouble recalling the nature and amount of medication taken in a specific period. Therefore, the healthcare providers believed that including an easy to use and intuitive medication reminder feature in the app would remind patients about previously ingested or impending medication.


*“Having that in the app, so they’re documenting it […] I think from a provider standpoint it would be great, but from the patient standpoint, we can’t get them to write it down in a book. It would have to be very simple. Like they go in and click a button or two, you know, have their medications, already populated and they could just go in and go click, click, click.”*

*—S07*


**Activity Reminder**: a few of the healthcare providers (37.5%; 3/8) suggested designing a feature that would help patients log their activities and remind them to exercise/stay active while giving the provider access to that information.


*“[…] we need something to help them exercise on here and way of recording it. [Even though] those are already with Fitbit’s and stuff, but that needs to be sent to the physician.”*

*—S04*


##### Information Communication and Presentation

Although the healthcare providers encouraged open lines of communication with their patients, they highlighted key features relating to data communication and presentation, such as synchronous and asynchronous communication and information presentation.

**Synchronous vs. Asynchronous Communication**: two providers (18.2%; 2/11) stressed the need for both providers and patients to be able to communicate and exchange information, even for a self-management tool.


*“Is there one-way or two-way communication with this app? It could be two-way. It has to be two. If it’s two-way, I’d feel comfortable. If it’s only one-way, it’s not worth it.”*


An example of a such one-way or asynchronous communication method is the use of voice notes. The providers had mixed feelings about the use of voice notes as a means of communicating with patients. While some healthcare providers (27.27%; 3/11) believed that voice notes can be beneficial to patients who were visually challenged, the majority (72.72%; 6/11) were against the use of voice notes, citing issues with understanding patient accents and dialects.


*“[…] for those with really poor eyesight, it’s gonna have to be a voice [recording], in their language.”*

*—S04*



*“But you know as well as I do, there’s so many dialects […] word accents. Sometimes you can’t understand.”*

*—S03*


The majority of providers (63.63%; 7/11) believed that sending text messages can be a useful medium to communicate specific, personalized, and urgent messages or instructions to patients, while a few preferred a chat feature.


*“[…] texts would be for urgent things like too high or too low [blood sugar].”*



*“[…] what I like and what I think a lot of the younger crowd would like, would be, that “chat.” […] You know, if you have questions, you’re gonna chat”*


**Information Presentation**: most providers were highly supportive (72.72%; 8/11) of having graphs in the mobile app. However, they remarked that some patients in their communities had challenges in comprehending the information provided in graphs and would often require the healthcare providers to describe it to them. The providers emphasized the need for the graphs to be simple, easy to read, with clearly displayed limits, and intuitive ways to warn patients about abnormal sugar levels.


*“[…] People respond visibly very easily using warning colors. Green, good, red, bad. The line where yours is. Pictures and graphs are great and probably better than texts.”*

*—S08*


Some healthcare providers indicated that adding appropriate pictures can help patients understand, interpret, and potentially maintain glycemic control.


*“[…] they see somebody happy; they know their blood sugars in a happy range. Uh, see some blood sugars, they, they maybe they can follow it on a chart day to day. Happy face here means they’re in control. A sad face here means they’re out of control.”*

*—S04*



*“[…] every picture tells a story. I think they would like pictures. See where they were and where they’re going.”*

*—S09*


Of the six healthcare providers who responded to this question, 50% (3/6) were cautiously optimistic about the use of tables and charts to communicate clinical data to patients, while the other half felt that the patients might be overwhelmed.

##### Patient-Related Challenges or Barriers

When asked about the potential to implement a diabetes self-management app in a clinical setting, healthcare providers highlighted key barriers that could impede care. Their responses were classified into the following subthemes: (a) patient literacy levels; (b) privacy concerns; and (c) lack of motivation.

**Patient Literacy Levels**: several providers (45.45%; 5/11) emphasized the diversity in education levels of the patients and questioned the patients’ ability to read and interpret graphs or other information on an application. Furthermore, providers raised concerns about the patients’ general readiness to use technology-based interventions, age-related usability barriers, and the language barriers.


*“Well, like I said, the people I’m going to use it on are usually older people and those people didn’t grow up with technology.”*

*—S10*



*“Most of my people speak Spanish or Spanglish.”*

*—S02*


Some providers (36.36%; 4/11) indicated that many of their patients can struggle with self-management, which can require them to visit the physician (in-person) to interpret their readings. This can in turn exacerbate issues related to access and geographical barriers. They suggested integrating the app with existing technologies, such as telemonitoring, or providing means of communicating relevant information to address this barrier.


*“If they have to come to the office […] to present the data, that’s a barrier. If it can be like the telehealth telemonitoring it’s transmitted and that’s not a barrier for them.”*

*—S04*



*“[…] Transportation is a big barrier to adopt something like this. [Because] they have to get to the office. They also have looking for rides and I’m in a neighborhood, lot of people walk to my place, well [those] people have to take a bus.”*

*—S02*


**Privacy Concerns**: a few providers (18.18%; 2/11) cited privacy concerns, highlighting that patients can be unwilling to be monitored or reluctant to share data.


*“I don’t know if they’d be open to doing something like that. […] most of them don’t […]. They don’t want something intruding on their […] autonomy I guess.”*

*—S10*


**Lack of Motivation**: a few providers (18.18%; 2/11) also indicated that some of the patients in underserved areas can lack motivation to adhere to their treatment. This concern in turn relates to sustained app use for effective adoption of a healthier lifestyle.

## 4. Discussion

Our qualitative investigation into the requirements for a diabetes self-management app provided rich data on the key features and functionalities for patient adoption and engagement. These data lend insights into the facilitators and barriers that can encourage or impede the patients’ sustained use of a diabetes self-management app. Although there has been research on the preferences of medically underserved patients [25,26], our study adds the multi-stakeholder perspective of providers in medically underserved communities.

Our findings are consistent with the previous literature on essential features for a diabetes self-management app [41,42,43]. Evidence-based guidelines suggest that logging and tracking blood glucose levels are essential elements in diabetes management [44,45]. Complementing these guidelines, Chavez and colleagues stated that physical activity, nutrition, blood glucose testing, medication or insulin dosage, health feedback, and education were key diabetes management tasks [46]. Consistent with this literature, patients in our study highlighted the need for logging and monitoring blood glucose levels, including visual trends for such values, tips about health lifestyle choices (e.g., exercise and nutrition), and reminders—all key basic features in a diabetes self-management app [42,43]. In addition, patients also requested the creation of a schedule feature, which would likely help them track their medication intake. Moreover, patients emphasized their preference to have a list of the types of foods they can consume to maintain glycemic control and to have a feature that can serve as a carbohydrate “tracker.” Research has shown that people with diabetes in low income and minority neighborhoods have limited access to healthy foods and limited discussions with healthcare providers about healthy eating [47]. In addition, it has been suggested that cultural factors, such as the preference for carbohydrate-rich foods, should be considered when understanding the prevalence of diabetes in Latinx/Hispanic communities in the U.S. [48]. Moreover, eating disorders are more prevalent in individuals with type 1 diabetes, in comparison to the general population [49]. Finally, while our findings did not include the desire for mental health support, previous work suggests that patients with diabetes support the inclusion of features for assistance with the psychological and emotional aspects of diabetes self-management, such as stress management and mechanisms to cope with negative emotions [43]. 

Patients were particularly interested in accessing educational resources to help them better self-manage their condition. Although patients interviewed in the study were part of a program that provided diabetes education, our results indicate their preference to have access to dietary tips and educational content in a mHealth app. Such a tool can remove the barriers related to access to educational resources and can serve as a central repository of educational information that patients can access on-demand with tailored content based on their preferences. The request for educational content was also supported by the healthcare providers in our sample, which highlights a potential gap and unmet need in existing apps, with education among the most underrepresented features [26,41]. Furthermore, it is important to note that content should be provided at levels consistent with the educational background of the population (primarily high school level or less in our sample). It is imperative that the content provided is aligned to recommendations in the literature to overcome the challenges and limitations related to the patients’ literacy levels. Design implications, such as minimizing technical jargon [50,51,52,53,54,55], presenting simplified language into tangible units [50,51,52,53], explaining uncommon terms [50,51,52,53], aligning content to the patients’ cultural background [50,51,52], and implementing visual and audible features over the use of text [50,51,52,53,54,55,56], must be considered when creating the educational content to be integrated in the mobile app. Williams and Schroeder [26] go on to state that the use of video-based educational material can complement text-based content since Hispanics are among the major consumers of online multimedia content [57]. Additionally, the availability of educational content in multiple languages seems to be essential to overcome language barriers, especially in those underserved communities in which patients with diabetes are predominantly non-English speakers. Finally, while participants mainly emphasized the need for educational content about healthy eating, further research is warranted to understand the perspectives about additional interventions to encourage behavioral change regarding eating habits. Participants also supported the integration of a fitness and physical activity module. However, given the availability and prevalence of apps for physical activity, further work is needed to understand the utilization of such apps among underserved populations and investigate unmet needs.

Our findings showed that patients were partial to understanding and interpreting their diabetic parameters through graphs and visualization. Although healthcare providers were generally supportive of these media, they cautioned against complex displays. Healthcare providers cautioned about the patients’ literacy levels and technical literacy levels, questioning their ability to read graphs and information in the app. Additionally, research suggests that the limited health literacy influence patients sustained motivation for engaging in monitoring their condition through self-management [58]. This can suggest that the design of health graphs and visualizations should account for the users’ literacy levels. This is in line with evidence suggesting the benefits to the patients’ health outcomes, when interventions sensitive to low health literacy limitations are used [30]. One of the healthcare providers suggested the use of colors and imagery to convey meaning and urgency, a view supported by the findings of Desai and colleagues, who suggested the use of a traffic light representation and facial emotions among possible visualizations of blood glucose forecasts [31]. Furthermore, while some patients indicated that they would benefit from the blood sugar forecasting capabilities in the app, research has shown that individuals with low numeracy can find it difficult to interpret the uncertainties associated with a forecast, consequently leading to disengagement [31]. The research suggests that visualizations that are simple, information-rich, convey authority, and promote actionable and learning behaviors from users are more effective in assisting users with data interpretation [31]. Additionally, providing the user with step-by-step guidance (e.g., a welcome wizard) about the different screen or features in the app can help address this problem [59]. Additionally, research has shown that difficulties using a smartphone can also impact the use of mobile apps [60]. Therefore, emphasis should be given towards orienting users to the app when they seem to be newer adopters of smartphone technology.

Patients expressed an interest in logging data and notes relating to their diabetes self-management. Healthcare providers also encouraged patients to log information, in the belief that it would keep patients engaged in their treatment. However, research has shown that a common reason for abandoning the use of apps is due the time required to enter data [61], which our providers also considered as an issue. Self-management practices can be encouraged by reducing the burden of data entry, for instance through simplified interfaces with adjustable text icon sizes to cater to individuals with visual impairments [26]. Additionally, data entry tasks could be simplified by the integration of databases and auto-fill features that help to minimize the amount of information the patients must recall and enter. Furthermore, providers highlighted the importance of synchronous and asynchronous communication capabilities; however, healthcare providers highlighted difficulties that can arise due to language barriers. Therefore, the availability of communication functions should integrate translators. Additionally, ways to effectively manage synchronous or patient-initiated communications must be explored to avoid unnecessary burden on providers, since concerns about potential changes in workload and overwhelming number of notifications have been highlighted in the literature regarding provider support of mHealth interventions for diabetes self-management [62]. 

In our study, patients with diabetes expressed their preference for simple instructions for healthy diets (i.e., a list of do’s and don’ts for healthy eating). This finding is consistent with the results of Turchioe et al. [32] that showed low-income diabetic patients’ mixed attitudes towards goal setting for dietary intake, and the general lack of positive attitudes towards personalized decision support and self-discovery. This is in contrast to findings in studies with more advantaged populations (e.g., [63]), where more detailed information on the impact of dietary consumption on glucose levels were preferred. These differences warrant further research to investigate adaptive or personalized interventions that take the user’s characteristic in mind and/or provide personalization capabilities for various user types. 

The synergistic interplay between remote patient monitoring (RPM) systems and smartphone apps can play a key role in assisted diabetes self-management. Patients and healthcare providers agreed that seamless integration of the app with their healthcare infrastructure can improve monitoring and managing diabetes. Studies have shown that RPM systems facilitate monitoring vital signs and allow for early detection of potentially hazardous health conditions, allowing time for provider intervention and preventing expensive hospital admissions [64]. For example, a clinician can track an insulin-dependent patient’s blood sugar profile, identify hypoglycemic tendencies, and make the necessary changes to the patient’s insulin dosages. In addition, a recent study demonstrated the potential of an RPM-facilitated diabetes management program, which incorporated evidence-based lifestyle interventions [65]. However, enthusiasm to adopt RPM in medically underserved communities should be tempered by the patients’ access to technology, user proficiency, and training requirements [66,67,68].

This work has certain limitations. First, our sample did not contain medically underserved patients beyond those residing in several counties in South Texas. Therefore, the results may not be generalizable to other medically-underserved communities. A similar limitation applies to the clinicians who were interviewed in our study. To overcome this limitation, we recommend carrying out the same study in multiple medically underserved communities across Texas and the U.S. Next, our sample consisted of only those patients who were seeking diabetes education and care through TAM-HST. It is conceivable that there are other patients with diabetes in that region who can have additional design requirements or have faced additional barriers not represented here. In addition, while in this study we identified patients by their diabetes type, no information was collected about usage of insulin, especially among patients with type 2 diabetes. Future work should investigate the differences between the requirements for users who do and do not use insulin. Next, our sample size for providers is small due to overall shortage of providers in the medically underserved counties and limited access. While we reached saturation with our current sample, future work should collect the perspectives of more providers in other underserved communities across Texas and the U.S. to validate and expand our findings. Finally, a significant proportion of the patients indicated that they were not knowledgeable about smartphone or self-management apps, thereby failing to provide design features. Although this is an unfortunate scenario, it presents an opportunity to educate patients about the capabilities of mHealth in assisting with diabetes self-management. It is imperative that these patients be included in formative usability tests to gather their perspective on the app design.

## 5. Conclusions

This paper highlights key features and functional requirements for the design of a diabetes self-management app tailored to the underserved community. Some of our findings are consistent with previous literature on essential features for a diabetes self-management app for the general population, such as including features to log and track blood glucose levels, physical activity, nutrition, and medication and insulin dosage, in addition to including reminders and educational content about healthy lifestyle choices [41,42,46]. In addition to these commonalities, with the published literature about the general guidelines for the design of self-management apps for diabetes, our review of relevant literature and interviews with patients and physicians in some representative underserved areas suggest that specific design requirements for the underserved can improve the adoption, usability, and sustainability of such interventions. Despite the prevalence of several self-management apps, the emergence of patient education as a desired feature suggests the need for designers to pay closer attention to the patients’ linguistic abilities and health literacy levels. Both the patients and providers also strongly desired the use of appropriate visualizations of diabetes data. In this regard, we recommend further investigations into the types of visualizations that would facilitate the easy interpretation of diabetes data. The use of simplified interfaces, adjustable icons, databases, and auto-fill features were identified to simplify information visualization, information recall, and data entry tasks. The use of formative training and technology exposure were identified to address issues of low experience with technology and low knowledge about mHealth capabilities, which can affect adoption and sustained engagement. While our data suggest that patients in underserved communities desire educational content about healthy lifestyle choices (e.g., nutrition and exercise), it is important that this content is presented in a way that is sensitive to their social and economic limitations, cultural background, and promotes a healthy attitudes towards eating. The results from our study also provided insights into perceived patient adoption barriers, including health literacy levels, motivation, and privacy concerns. To mitigate these barriers, we recommend adopting a community-based participatory research approach to facilitate a grassroots-level education about the capabilities of the app being designed.

## Figures and Tables

**Table 1 ijerph-19-00127-t001:** Demographic information of patients.

Characteristic	Number of Respondents	Percentage
**Gender (*n* = 97)**		
Female	71	73.20
Male	26	26.80
**Income (*n* = 97)**		
Less than USD 20,000	30	30.93
USD 20,000–USD 30,000	19	19.59
USD 30,000–USD 40,000	11	11.34
USD 40,000–USD 50,000	10	10.31
USD 50,000–USD 60,000	4	4.12
Above USD 60,000	8	8.25
Prefer not to answer	15	15.46
**Race (*n* = 97)**		
White (non-Hispanic or Latinx)	7	7.22
Hispanic or Latinx (White)	61	62.89
Hispanic or Latinx (non-White)	26	26.80
American Indian or Native	2	2.06
Two or more races	1	1.03
**Education (*n* = 97)**		
Less than high school diploma	16	16.50
High school diploma or GED	29	29.90
Some college, no degree	26	26.80
Associate degree	14	14.43
Bachelor’s degree	9	9.28
Graduate or professional degree	3	3.09
**Type of Diabetes (*n* = 97)**		
Pre-diabetes	4	4.12
Type 1	9	9.28
Type 2	79	81.44
Do not know	5	5.16
**First Diagnosed with Diabetes (*n* = 97)**		
Less than 6 months	12	12.37
6 months to 1 year	12	12.37
Greater than 1 year to 10 years	35	36.09
Greater than 10 years to 20 years	32	32.99
Greater than 20 years	6	6.18

**Table 2 ijerph-19-00127-t002:** Demographic information of healthcare providers.

Characteristic	Number of Respondents	Percentage
**Gender (*n* = 11)**		
Female	2	18.19
Male	9	81.81
**Age (*n* = 11)**		
45–54 years	1	9.09
55–64 years	5	45.45
65–74 years	5	45.45
**Race (*n* = 11)**		
White (Non-Hispanic or Latinx)	9	81.81
Hispanic or Latinx (non-White)	2	18.19
**Nature of Experience (*n* = 11)**		
Family medicine/practice	9	81.81
General medicine	1	9.09
Pediatric nurse practitioner	1	9.09

## Data Availability

The data that support the findings of this study are available from the corresponding author, FS, upon reasonable request.

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
