# Peer review of "Eliciting Requirements for a Diabetes Self-Management Application for Underserved Populations: A Multi-Stakeholder Analysis"

_ijerph, 2021, doi:10.3390/ijerph19010127_

Round 1

Reviewer 1 Report

This study provides insight into the needs and preferences of an underserved community of people with diabetes living in Texas. The study authors have provided key findings from a large amount of qualitative data (97 interviews with patients, on average 45 minutes; 11 interviews with clinicians). This is a well thought out study, my comments below will hopefully help reflect on some of the findings, improve clarity, and link more closely into the existing literature. In general, more detail for the results section is also needed.

  1. Consider including the COREQ checklist as an appendix
  2. The overview of the literature in the introduction misses some key points. I list some additional references below that you may find useful. Please also note that one of Mamykina + team's papers is mentioned in the discussion but given that program of research’s high relevance to the current paper, this should be mentioned in the introduction as well.

Kim SH, Lee A. Health-literacy-sensitive diabetes self-management interventions: A systematic review and meta-analysis. Worldviews Evid Based Nurs. 2016;13(4):324-33. Epub 2016/04/23. doi: 10.1111/wvn.12157. PubMed PMID: 27104337.

Parker, S., Prince, A., Thomas, L., Song, H., Milosevic, D., Harris, M. F., & Group, I. S. (2018). Electronic, mobile and telehealth tools for vulnerable patients with chronic disease: a systematic review and realist synthesis. BMJ Open, 8(8), e019192. https://doi.org/10.1136/bmjopen-2017-019192

Turchioe MR, Heitkemper EM, Lor M, Burgermaster M, Mamykina L. Designing for engagement with self-monitoring: a user-centered approach with low-income, Latino adults with Type 2 Diabetes. International Journal of Medical Informatics. 2019. doi: https://doi.org/10.1016/j.ijmedinf.2019.08.001.

Caburnay CA, Graff K, Harris JK, McQueen A, Smith M, Fairchild M, et al. Evaluating diabetes mobile applications for health literate designs and functionality, 2014. Prev Chronic Dis. 2015;12:E61. doi: http://dx.doi.org/10.5888/pcd12.140433. PubMed PMID: 25950568; PubMed Central PMCID: PMCPMC4436041.

Krebs P, Duncan DT. Health app use among US mobile phone owners: A national survey. JMIR Mhealth Uhealth. 2015;3(4):e101. DOI: 10.2196/mhealth.4924.

  1. Consider a stronger public health/equity focus – you can mention the digital divide and refer to the literature briefly. There is also research indicating that people who are older/lower eHealth literacy tend to be less likely to use diabetes apps (see Krebs 2015, above)
  2. The ‘recent advances’ paragraph would benefit from greater detail about technologies for facilitating self management, including discussion of continuous blood glucose monitoring /other wearables and associated apps, machine learning algorithms, and in general, systematic reviews on the effectiveness of mHealth technologies for diabetes self-management.

Methods

  1. Give dates of recruitment
  2. Can you describe the diabetes conference (recruitment site) in more detail? + maybe more about context of what is already available to these patients (e.g. via TAM HST)
  3. Please describe findings from references 21 and 22 in greater detail
  4. Did you collect data on how recently the person was diagnosed? This would also be useful for context.
  5. Methods needs more detail about what is included in the interview schedule in the methods section.

Results:

  1. What proportion of patients were using insulin? Recommendations for monitoring blood glucose may be different for those using and not using insulin.
  2. Really interesting that only 17% discussed the need to understand what blood glucose entries mean or why they were high/low. This discussion should consider findings in relation to the Mamykina papers in greater depth.
  3. Interesting that participants were focused on trends rather than the effects of individual foods, or exercise, for example. Qualitative research on diabetes apps shows that this is also often desired. Can you comment on this and either revise the results or add commentary to the discussion.
  4. Please improve the clarity around ‘foods to consume to avoid hypoglycaemia’ in the section on diet regulation. Did you have participants who mentioned issues with foods that made blood glucose levels higher? Consuming foods to treat hypoglycaemia is different to ‘diet.’
  5. Generally the themes would benefit from greater detail e.g. for participants who wanted help with ‘diet regulation’ is this mostly around education? Or did they want help creating habits? Navigating social events? It is interesting that in the quotes this mainly looks like participants want black and white rules (Do’s and Do nots). How does this findings fit into other diabetes literature? Can you elaborate on the fitness and physical activity theme? What benefit did they perceive from integrating it into a diabetes app? What aspects of ‘fitness’ did they want to monitor? Was there anything that the currently available apps can’t do that would be a useful feature? Had participants had experiences with apps that had poor usability or were invasive? What does easy to use look like for these participants? E.g was there any discussion of overwhelming vs manageable amounts of information? This would complement the themes identified by clinicians
  6. Synchronous communication – why was the participant quoted not happy with one way communication?
  7. It is unclear from the results whether participants are discussing messages that would be automated or sent by a doctor/nurse. Is this available from the quotes? Elsewhere primary care physicians have commented that they did not want to feel ‘on call’ with their patients with regards to notifications (see e.g. Ayre et al 2019 JMIR Mhealth UHealth: https://mhealth.jmir.org/2019/1/e11885/ )
  8. The ‘temporal and physical barriers’ subheading is missing

Discussion

  1. Can you elaborate on the educational resources patients were already receiving and what the benefits would be of receiving similar/overlapping content via an app?
  2. Please consider incorporating a discussion of healthy attitudes towards eating as this is an important aspect of diabetes self-management – this can draw on the finding about wanting very black and white rules about foods.
  3. Did you ask what apps people were already using? If not this and other contextual information (insulin use, years since diagnosed) may be worth listing as a limitation
  4. Please also comment on having relatively fewer GPs (n=11) in the limitations and what this means when interpreting results from the two separate samples.
  5. Fundamentally, after reading this article I was left wondering to what extent were their needs and preferences in this sample different to what you would expect from a more advantaged sample? This needs to be really clear through thoughtful engagement with the existing literature. It is fine if some things are the same and others are different, but making this very clear will help progress this area of research and highlight the clear importance of engaging with disadvantaged populations when designing diabetes apps.

Author Response

We would like to sincerely thank the the reviewer for their comments and positive feedback. Below we have detailed our responses to each comment along with direct quotes or pointers to relevant sections in the manuscript.

Comment 1.1. Consider including the COREQ checklist as an appendix

Response 1.1. Thanks for the comments. The COREQ checklist has been attached to the submission.

Comment 1.2. The overview of the literature in the introduction misses some key points. I list some additional references below that you may find useful. Please also note that one of Mamykina + team's papers is mentioned in the discussion but given that program of research’s high relevance to the current paper, this should be mentioned in the introduction as well.

Response 1.2. Thanks for recommending these relevant papers. The recommended papers have been reviewed, and the introduction (3rd-5th paragraphs) has been expanded to address the reviewer’s concern.

Comment 1.3. Consider a stronger public health/equity focus – you can mention the digital divide and refer to the literature briefly. There is also research indicating that people who are older/lower eHealth literacy tend to be less likely to use diabetes apps (see Krebs 2015, above)

Response 1.3. Thanks for the comment. Content has been added to the 4th paragraph of the introduction to emphasize literacy-, age-, and financial-related key points, which provides a stronger health/equity focus.

Comment 1.4. The ‘recent advances’ paragraph would benefit from greater detail about technologies for facilitating self-management, including discussion of continuous blood glucose monitoring /other wearables and associated apps, machine learning algorithms, and in general, systematic reviews on the effectiveness of mHealth technologies for diabetes self-management.

Response 1.4. Content has been added to the referenced paragraph to address the issues indicated.

Methods:

Comment 1.5. Give dates of recruitment

Response 1.5. The following sentence (1st paragraph of the Methods section) has been edited to address this comment:

Interviews with patients were conducted by four nurse educators with graduate degrees in nursing or education during diabetes self-management education sessions held between April 8, 2019 and May 3, 2019, as part of the Healthy Texas initiative.

Comment 1.6. Can you describe the diabetes conference (recruitment site) in more detail? + maybe more about context of what is already available to these patients (e.g. via TAM HST)

Response 1.6. The following sentences were added (1st paragraph of the Methods section) to address this comment:

These education sessions aimed at educating patients with diabetes on practical strategies and tips for incorporating healthy behaviors in their daily activities including effective nutrition, general health and wellness, the role of physical activity, and ways to mitigate the financial and physical burden of diabetes

Comment 1.7. Please describe findings from references 21 and 22 in greater detail

Response 1.7. As discussed above (in response to Comments 1.2 and 1.3), findings from references 21 and 22 have been discussed in greater detail in the Introduction.

Comment 1.8. Did you collect data on how recently the person was diagnosed? This would also be useful for context.

Response 1.8. Thanks for this comment. A new section (First Diagnosed with Diabetes) has been added to Table 1 to provide information about this matter.

Comment 1.9. Methods needs more detail about what is included in the interview schedule in the methods section.

Response 1.9. Thanks for the recommendation. The following paragraph (2nd paragraph of the Methods section) was expanded to address this comment:

Two interview protocols were developed for patients and providers, respectively, to reduce individual biases and assumptions and to standardize the interviews. Interviews with patients focused on understanding their expectations from a diabetes self-management app. The questions in the interview protocol for patients included topics such as perceived barriers and limitations for diabetes self-management, the use of technology to manage diabetes, important characteristics in a technology for diabetes self-management, and preferences on features for an app for diabetes self-management. Similarly, interviews with providers focused on their expectations from a self-management app for diabetes both from the patients’ perspective and the type of information or interactions providers expected from such a tool. The questions in the interview protocol for providers included topics such as perceived barriers and limitations for patients to adopt and app for diabetes self-management, perceived barriers and limitations for providers to monitor patient who have adopted such technology, perceived importance on feature for an app for diabetes self-management, and preferences about data representation and data communication. The interviews took approximately 45 minutes for both patients and providers. Patients and providers received a $25 or $50 gift card, respectively, for participation. The Texas A&M University Review Board reviewed and approved this study (IRB Protocol #IRB2018-1503D) and all participants provided informed consent.

Results:

Comment 1.10. What proportion of patients were using insulin? Recommendations for monitoring blood glucose may be different for those using and not using insulin.

Response 1.10. Unfortunately, this information was not collected. The following was added to the limitation section:

In addition, while in this study we identified patients by their diabetes type, no information was collected about usage of insulin, especially among patients with Type-2 diabetes. Future work should investigate the differences between requirements for users who do and do not use insulin.

Comment 1.11. Really interesting that only 17% discussed the need to understand what blood glucose entries mean or why they were high/low. This discussion should consider findings in relation to the Mamykina papers in greater depth.

Response 1.11. Thanks for the comments. Content has been added to the discussion of visualizations and graphs (4th paragraph in the Discussion section) to integrate Mamykina’s findings regarding the use of effective visualizations to support data interpretation.

Research suggests that visualizations that are simple, information-rich, convey authority, and promote actionable and learning behaviors from users are more effective in assisting users with data interpretation [31].

Comment 1.12. Interesting that participants were focused on trends rather than the effects of individual foods, or exercise, for example. Qualitative research on diabetes apps shows that this is also often desired. Can you comment on this and either revise the results or add commentary to the discussion?

Response 1.12. We believe this is somewhat addressed in the 2nd paragraph of the Discussion section as follows. We’d be happy to integrate specific qualitative research the reviewer is referring to if the reviewer kindly provides the citations.

Our findings are consistent with previous literature on essential features for a di-abetes self-management app [41,42]. Evidence-based guidelines suggest that logging and tracking blood glucose levels are essential elements in diabetes management [43,44]. Complementing these guidelines, Chavez and colleagues stated that physical activity, nutrition, blood glucose testing, medication or insulin dosage, health feed-back, and education were key diabetes management tasks [45]. Consistent with this lit-erature, patients in our study highlighted the need for logging and monitoring blood glucose levels including visual trends for such values, tips about health lifestyle choic-es (e.g., exercise, nutrition), and reminders—all key basic features in a diabetes self-management app [42]. In addition, patients also requested the creation of a schedule feature, which would likely help them track their medication intake.”

Comment 1.13. Please improve the clarity around ‘foods to consume to avoid hypoglycaemia’ in the section on diet regulation. Did you have participants who mentioned issues with foods that made blood glucose levels higher? Consuming foods to treat hypoglycaemia is different to ‘diet.’

Response 1.13. Thanks for pointing this out. We believe this example may be confusing and since we do not have more specific quotes from participants to expand, we removed it to avoid confusion.

Comment 1.14.Generally, the themes would benefit from greater detail.

Response 1.14. While these are great suggestions, unfortunately after reviewing the data, we do not have additional anecdotal details to add. However, most of these issues were added to the discussion section in response to your comments on discussion, and the need for future work was motivated accordingly.

Comment 1.14.1. e.g. for participants who wanted help with ‘diet regulation’ is this mostly around education? Or did they want help creating habits? Navigating social events? It is interesting that in the quotes this mainly looks like participants want black and white rules (Do’s and Do nots). How does these findings fit into other diabetes literature?

Response 1.14.1. This is a great point. While we don’t have data to expand on the results, we added the following to the 2nd paragraph of the discussion:

Although in our study patients with diabetes expressed their preference for simple instructions for healthy diets (i.e., normative instructions or do’s and don’ts list for healthy eating), an effective mHealth interventions for diabetes self-management in underserved communities must account for social and economic sensitivities and users’ cultural backgrounds, while promoting healthy-eating attitudes. Finally, while participants mainly emphasized the need for educational content about healthy eating, further research is warranted to understand the perspectives about additional interventions to encourage behavioral change regarding eating habits.

Comment 1.14.2. Can you elaborate on the fitness and physical activity theme? What benefit did they perceive from integrating it into a diabetes app? What aspects of ‘fitness’ did they want to monitor? Was there anything that the currently available apps can’t do that would be a useful feature?

Response 1.14.2. To address this comment, we added the following to the 2nd paragraph of the Discussion section.

Participants also supported the integration of a fitness and physical activity module. However, given the availability and prevalence of apps for physical activity, further work is needed to understand the utilization of such apps among underserved populations and investigate unmet needs.

Comment 1.14.3. Had participants had experiences with apps that had poor usability or were invasive? What does easy to use look like for these participants? E.g was there any discussion of overwhelming vs manageable amounts of information? This would complement the themes identified by clinicians.

Response 1.14.3 While we asked participants about their experience with other diabetes self-management apps, only two has such experience. Therefore, we opted to not include it in the manuscript.

Comment 1.15. Synchronous communication – why was the participant quoted not happy with one way communication?

Response 1.15. Thank you for the comment. We believe a potential reason is included in the Asynchronous Communication section. To address this, we merged the two sections to improve the flow.

Comment 1.16. It is unclear from the results whether participants are discussing messages that would be automated or sent by a doctor/nurse. Is this available from the quotes? Elsewhere primary care physicians have commented that they did not want to feel ‘on call’ with their patients with regards to notifications (see e.g. Ayre et al 2019 JMIR Mhealth UHealth: https://mhealth.jmir.org/2019/1/e11885/)

Response 1.16. The messages are in the context of the doctors’ sending messages to patients outside of the clinic. There was no mention of “automated” messages. The text was modified to include “sending” in the body to clarify that the messages are sent manually and not from an automated system.

Also, thank you for sharing the Ayre et al. citation. We added the following to 5th paragraph in the Discussion section:

Additionally, ways to effectively manage synchronous or patient-initiated communications must be explored to avoid unnecessary burden on providers, since concerns about potential changes in workload and overwhelming number of notifications have been highlighted in the literature regarding providers support of mHealth interventions for diabetes self-management [60].

Comment 1.17. The ‘temporal and physical barriers’ subheading is missing.

Response 1.17. Thanks for pointing this out. The section should only include the three subthemes discussed, therefore ‘temporal and physical barriers’ has been removed and the manuscript has been edited accordingly.

Discussion:

Comment 1.18. Can you elaborate on the educational resources’ patients were already receiving and what the benefits would be of receiving similar/overlapping content via an app?

Response 1.18. The educational resources’ patients already received were added to the Methods section in response to Comment 1.6. In addition, new content has been added (3rd paragraph in the Discussion section) to further discuss the benefits of receiving similar/overlapping content via mobile apps.

Comment 1.19. Please consider incorporating a discussion of healthy attitudes towards eating as this is an important aspect of diabetes self-management – this can draw on the finding about wanting very black and white rules about foods.

Response 1.19. Thanks for the comment. Content has been added to the 2nd paragraph of the Discussion secion to discuss factors to consider to integrate nutritional content that promotes healthy attitudes towards eating.

Also, patients emphasized their preference to have a list of the types of foods they could consume to maintain glycemic control and to have a feature that could serve as a carbohydrate “tracker.” Research has shown that people with diabetes in low income and minority neighborhoods have limited access to healthy foods and limited discussions with healthcare providers about healthy eating [Breland et al., 2013]. In addition, it has been suggested that cultural factors, such as preference for carbohydrate-rich foods should be considered when understanding the prevalence of diabetes in Latinx/Hispanic communities in the U.S. [Aguayo-Mazzucato, 2018]. In addition, research has shown that eating disorders are more prevalent in individuals with Type 1 diabetes, in comparison to the general population [Colton et al., 2009]. Although in our study patients with diabetes expressed their preference for simple instructions for healthy diets (i.e., do’s and don’ts list for healthy eating), a effective mHealth interventions for diabetes self-management in underserved communities must account for social and economic sensitivities, users’ cultural backgrounds, while promoting healthy-eating attitudes.”

Comment 1.20. Did you ask what apps people were already using? If not this and other contextual information (insulin use, years since diagnosed) may be worth listing as a limitation

Response 1.20. As mentioned above, time since diagnoses was added to Table 1. However, information about insulin usage and current/past app usage was not included and were discussed as a limitation.

Comment 1.21. Please also comment on having relatively fewer GPs (n=11) in the limitations and what this means when interpreting results from the two separate samples.

Response 1.21. Thanks for the comment. Content has been added to the limitations section to address this issue.

Next, our sample size for providers is small due to overall shortage of providers in the medically underserved counties and limited access. While we reached saturation with our current sample, future work should collect the perspectives of more providers in other underserved communities across Texas and the U.S. to validate and expand our findings.

Comment 1.22. Fundamentally, after reading this article I was left wondering to what extent were their needs and preferences in this sample different to what you would expect from a more advantaged sample? This needs to be really clear through thoughtful engagement with the existing literature. It is fine if some things are the same and others are different, but making this very clear will help progress this area of research and highlight the clear importance of engaging with disadvantaged populations when designing diabetes apps.

Response 1.22. We wholeheartedly agree with this comment. We believe that the new and significantly expanded introduction would somewhat address this comment. To further address this comment, we revised the conclusion to highlight this important issue.

Reviewer 2 Report

Dear authors, thank you for submitting the manuscript to the International Journal of Environmental Research and Public Health.

This paper highlights key features and functional requirements for the design of a diabetes self-management app tailored to the underserved community. The authors carried out semi-structured survey interviews with 97 patients with diabetes and 11 healthcare providers to elicit perspectives and preferences regarding a diabetes self-management app, and their beliefs regarding such an app’s usage and utility.

The topic is extremely interesting and may have relevant policy implications, both at the national and international educational levels. I consider the paper connected with the overall philosophy of the Journal.

The paper is generally well-written. The authors have clearly done a lot of work and their research design is set out clearly.

My suggestion for improvement is related to the fact that the work is complex and may seem complicated to study and understand. Thus, several tables and graphs can be created through which the answers to the questions specified in the questionnaires can be reproduced.

Author Response

Thank you for your positive and supportive feedback. Also, thank you for your suggestion to present the results in table/graph format. Given the extent of results provided and the formatting guidelines, we will discuss this with the associate editor and revise accordingly. 

Round 2

Reviewer 1 Report

Thank you for addressing the comments, I know they were quite extensive! There are a few remaining aspects of the existing literature that I would encourage you to consider:

  1. Thank you for including Turchioe et al 2019. One of the key findings from this study was that "attitudes were more mixed towards goal setting, and very few participants responded positively to self-discovery and personalized decision support features." This concept is really relevant to your findings as you've also seen that participants were interested in overall trends (and do/do not rules), and less interested in learning about specific impacts of various foods. I think this would be worthwhile to draw out in the discussion section.
  2. The above (#1) contrasts with research (in more advantaged populations) showing that e.g. paired glucose testing, to demonstrate the impacts of different foods on blood sugar levels, is actually quite effective and engaging for users. You can look to Hartmann-Boyce, 2018 (https://doi.org/10.1177/1049732318784815), and Tanenbaum 2015 ( https://doi.org/10.1111/dme.12745 ) for good summaries of this research. You can refer to Ayre 2021 (https://doi.org/10.1080/08870446.2021.1909023) as an example of research looking at the role of health literacy in self-monitoring for diabetes self-management.
  3. Please also consider including Baptista 2019 article (https://www.ncbi.nlm.nih.gov/pubmed/31166804) - this is an Australian survey that asked >300 people with type 2 diabetes about their needs in a diabetes app. This also identified support for psychological and emotional support. This should be compared and contrasted to findings from the current study. 

A lot of existing literature, I know! I do think that taking the time to consider existing work and reflecting on this in the discussion will substantially strengthen the paper. At the moment the sections of the discussion that do this are quite long and the key points (and implications) are not too clear - please revise and you could perhaps include a table to help summarise. 

Author Response

Comment 1.1. Thank you for including Turchioe et al 2019. One of the key findings from this study was that "attitudes were more mixed towards goal setting, and very few participants responded positively to self-discovery and personalized decision support features." This concept is really relevant to your findings as you've also seen that participants were interested in overall trends (and do/do not rules), and less interested in learning about specific impacts of various foods. I think this would be worthwhile to draw out in the discussion section.

Response 1.1. Thank you for this suggestion. We added a paragraph to our discussion to address this comment, as follows:

In our study patients with diabetes expressed their preference for simple instructions for healthy diets (i.e., a list of do’s and don’ts for healthy eating). This finding is consistent with Turchioe et al.’s [32] results that showed low-income diabetic patients’ mixed attitudes towards goal setting for dietary intake and the general lack of positive attitudes towards personalized decision support and self-discovery. This is in contrast to findings in studies with more advantaged populations [e.g. 50], where more detailed information on the impact of dietary consumption on glucose levels were preferred.  These differences warrant further research to investigate adaptive or personalized interventions that take user’s characteristic in mind and/or provide personalization capabilities for various user types.”     

Comment 1.2. The above (#1) contrasts with research (in more advantaged populations) showing that e.g. paired glucose testing, to demonstrate the impacts of different foods on blood sugar levels, is actually quite effective and engaging for users. You can look to Hartmann-Boyce, 2018 (https://doi.org/10.1177/1049732318784815), and Tanenbaum 2015 (https://doi.org/10.1111/dme.12745) for good summaries of this research. You can refer to Ayre 2021 (https://doi.org/10.1080/08870446.2021.1909023) as an example of research looking at the role of health literacy in self-monitoring for diabetes self-management.

Response 1.2. This great comment is also addressed in the new paragraph added in response to Comment 1.1. We had some difficulties in using Hartmann-Boyce work since it was not done in the context of diabetes. We also used Ayre et al. in the fourth paragraph of the Discussion section to support the argument for the importance of health literacy.

Comment 1.3. Please also consider including Baptista 2019 article (https://www.ncbi.nlm.nih.gov/pubmed/31166804) - this is an Australian survey that asked >300 people with type 2 diabetes about their needs in a diabetes app. This also identified support for psychological and emotional support. This should be compared and contrasted to findings from the current study. 

Response 1.3. We added this to the second paragraph of the Discussion section as follows:

“…Finally, while our findings did not include desires for mental health support, previous work suggests that  patients with diabetes support the inclusion of features for assistance with psychological and emotional aspects of diabetes self-management, such as stress management and mechanisms to cope with negative emotions [43].”

Comment 1.4. A lot of existing literature, I know! I do think that taking the time to consider existing work and reflecting on this in the discussion will substantially strengthen the paper. At the moment the sections of the discussion that do this are quite long and the key points (and implications) are not too clear - please revise and you could perhaps include a table to help summarize.

Response 1.4. Thank you so much your thorough comments and positive feedback. We revised the discussion throughout and made organizational changes to improve the flow and to highlight key findings/discussion points better. We hope with this new re-organization that clearly identifies a key discussion point per paragraph, a table would not be needed.